# The Marine Dinoflagellate *Alexandrium minutum* Activates a Mitophagic Pathway in Human Lung Cancer Cells

**DOI:** 10.3390/md16120502

**Published:** 2018-12-12

**Authors:** Christian Galasso, Genoveffa Nuzzo, Christophe Brunet, Adrianna Ianora, Angela Sardo, Angelo Fontana, Clementina Sansone

**Affiliations:** 1Stazione Zoologica Anton Dohrn, Istituo Nazionale di Biologia, Ecologia e Biotecnologie Marine, Villa Comunale, 80121 Naples, Italy; adrianna.ianora@szn.it (A.I.); clementina.sansone@szn.it (C.S.); 2Bio-Organic Chemistry Unit, Institute of Biomolecular Chemistry-CNR, Via Campi Flegrei 34, Pozzuoli, 80078 Naples, Italy; nuzzo.genoveffa@icb.cnr.it (G.N.); angela.sardo@icb.cnr.it (A.S.); a.fontana@icb.cnr.it (A.F.)

**Keywords:** glycoprotein, mitophagy, marine antiproliferative compounds, *Alexandrium minutum*

## Abstract

Marine dinoflagellates are a valuable source of bioactive molecules. Many species produce cytotoxic compounds and some of these compounds have also been investigated for their anticancer potential. Here, we report the first investigation of the toxic dinoflagellate *Alexandrium minutum* as source of water-soluble compounds with antiproliferative activity against human lung cancer cells. A multi-step enrichment of the phenol–water extract yielded a bioactive fraction with specific antiproliferative effect (IC_50_ = 0.4 µg·mL^−1^) against the human lung adenocarcinoma cells (A549 cell line). Preliminary characterization of this material suggested the presence of glycoprotein with molecular weight above 20 kDa. Interestingly, this fraction did not exhibit any cytotoxicity against human normal lung fibroblasts (WI38). Differential gene expression analysis in A549 cancer cells suggested that the active fraction induces specific cell death, triggered by mitochondrial autophagy (mitophagy). In agreement with the cell viability results, gene expression data also showed that no mitophagic event was activated in normal cells WI38.

## 1. Introduction

To date, many microalgae have been shown to inhibit proliferation and development of malignant cells. For example, the diatoms *Attheya longicornis*, *Chaetoceros socialis*, *Chaetoceros furcellatus*, *Skeletonema marinoi* and *Porosira glacialis* exhibited anti-cancer activity against melanoma A2058 cells [1]. An ethyl acetate crude extract from the diatom *Chaetoceros calcitrans* was able to induce apoptosis in breast cancer cells [2]. Aldehydes from *S. marinoi* induced apoptosis in colorectal Caco-2 tumor cells [3] and mes-c-myc A1 cell line [4], through an extrinsic apoptotic pathway [5]. Interestingly, the potential microalgal bioactivities can be modulated by culture conditions [6], highlighting that the synthesis of secondary metabolites responsible for such biological effects is an adaptive response to environmental cues. These molecules are probably synthetized to protect themselves and/or to reinforce responses to environmental stimuli, through activation of specific molecular pathways [7]. In addition, carotenoids from marine sources have been reported for their anti-proliferative effects [8]. From a study on the green alga *Dunaliella tertiolecta*, the carotenoid violaxanthin showed a potent antiproliferative activity on MCF-7 breast cancer cells and induced biochemical changes typical of early apoptosis [9]. Another carotenoid, peridinin, less known because uniquely present in some dinoflagellate species, induces apoptosis in DLD-1 human colon cancer cells [10].

Dinoflagellates represent a promising marine source, with more than 2000 species, covering a large biodiversity of ecological strategies. They can be free living, symbiotic, parasitic or mixotrophic. Most of them are known for the potent neurotoxins that they produce causing ecosystem and human health problems [11]. Human diseases associated with exposure to marine dinoflagellate toxins include paralytic, diarrheic, neurotoxic and ciguatera fish poisoning [12]. The toxins inducing these pathologies are chemically diverse and include macrolides, cyclic polyethers, spirolides and purine alkaloids [13]. In previous studies, some compounds produced by dinoflagellates showed interesting and specific in vitro antiproliferative effects against various cancer cell lines [14,15,16]. Species of the genus *Alexandrium* present peculiar features from this point of view [17]. Indeed, *Alexandrium pseudogonyaulax* excretes antimicrobial and antifungal substances such as goniodomin-A [18], which is also able to inhibit angiogenesis [19]. *Alexandrium ostenfeldii* produces the cyclic imine toxin 13-desmethyl spirolide C [20], a polyketide recently discovered for its anti-Alzheimer’s activity, being able to cross the blood–brain barrier in mice targeting nicotinic receptors [21].

These results obtained from *Alexandrium* species give strong impulse to screen this promising group of marine dinoflagellates, analyzing the chemical diversity of their secondary metabolites and their potential bioactivities and applications for human health.

We investigated the biological activity of extracts from *Alexandrium minutum*, known for toxin production affecting bivalves (e.g., oysters) [22]. Through a bioassay-guided fractionation, we isolated from a polar extract of this species a bioactive fraction with a specific antiproliferative effect on the human lung adenocarcinoma cells (A549), without affecting cell viability in human normal lung fibroblasts (Figure 3). Chemical analysis suggested that the hydrophilic high molecular weight molecule likely responsible for the activity was a glycoprotein. We also discovered that the antiproliferative effect in A549 cancer cells of this molecule is linked to the mitochondrial autophagy cell death pathway. Autophagy is an evolutionarily conserved catabolic process, aiming to the maintenance of cellular homeostasis and correcting functioning of intracellular organelles [23]. Mitochondrial are fundamental intracellular organelles responsible of the regulation of cellular homeostasis and cell death [24]; thus, the removal of damaged mitochondria is critical for maintaining proper cellular functions and viability. Mitochondria with damages or dysfunction are removed through a specific autophagic process called “mitophagy”. Mitophagy has been described in yeast as a process mediated by autophagy-related 32 gene (Atg32) and in mammals by NIP3-like protein X (NIX; also known as BNIP3L (BCL2 Interacting Protein 3 Like)) [25]. Mitophagy is regulated in many metazoan cell types by Parkin and PTEN-induced putative kinase protein 1 (PINK1), while mutations in these genes are linked to Parkinson’s disease [26,27,28].

To our knowledge, this is the first report of cell death activation triggered by specific mitophagic pathway induced by marine dinoflagellate derived compound, without affecting normal cells.

## 2. Results

### 2.1. Chemical Analyses

Extraction of *A. minutum* biomass (5.7 g wet weight) with TRI reagent^®^ gave 350 mg water soluble extract. Fractionation of this material by HR-X column led to four enriched fractions that were tested against human lung adenocarcinoma cells A549 (see Appendix A). Fraction 1B (5.3 mg), eluted with ACN/H_2_O 7:3, was the only one that showed cytotoxic effect, with an IC_50_ of 1.3 µg·mL^−1^. This fraction was also tested on a panel of human cells. Fraction 1B exhibited stronger cytotoxic effect on A549, with respect to human colorectal adenocarcinoma cells (HT29) and human prostate cancer cells (PC3). Moreover, the same fraction did not show cytotoxic effect on human normal lung fibroblasts (WI38) (see Appendix A). Diffusion NMR experiments are used to determine the size of macromolecules and aggregates according to their diffusion coefficients in solution. The spectra of the active Fraction 1B confirmed the presence of a family of macromolecules with different molecular weights. On the other hand, ^1^H NMR experiment of this fraction was characterized by signals between 3 and 5 ppm that were in agreement with a predominance of carbohydrates (Figure 1). After further fractionation by sequential ultrafiltration over membranes with cut-off of 3 kDa and 10 kDa, the activity was retained in the fraction above 10 kDa (Fraction 3B 0.7 mg). Significantly, this fraction (IC_50_ of 0.4 µg·mL^−1^) was four times more potent than the parent Fraction 1B. Further ultrafiltration over exclusion membranes led an enrichment of the cytotoxic above 30 kDa but the activity was also present in the filtrate.

Electrophoresis gel corroborated the co-presence of three major proteins in the active Fraction 3B of *A. minutum*. According to the intensity of the band stained by silver nitrate, two of these proteins accounted for minor components with a molecular weight higher than 50 kDa even if the main band was around 20 kDa. Analysis by colorimetric phenol–sulforic acid method [29] and Bradford assay [30] indicated that almost 96% of this sample was composed of carbohydrates and only 4% of protein. Hydrolysis of this material by acid treatment with a solution 2 M of Trifluoroacetic acid (TFA) led to a complete loss of the activity against A549 cells (data not shown). Analysis of the hydrolyzed sugars by high-performance anion-exchange chromatography (HPAEC) supported a large presence of D-galactose and D-glucose (63% and 37%, respectively) (see Appendix A).

### 2.2. Bioassay and Mechanism of Action

The activity of the *A. minutum* fractions was tested on human lung adenocarcinoma cells. Fraction 1B exhibited a strong cytotoxicity on A549 cells, with an IC_50_ = 1.3 µg·mL^−1^. The following steps of fractionation still enhanced the activity, also lowering the IC_50_. In particular, Fraction 2B presented an IC_50_ = 0.8 µg·mL^−1^ (Appendix A), while Fraction 3B reached an IC_50_ = 0.4 µg·mL^−1^ (Figure 2A). Fraction 3A did not significantly affect cell viability for all concentrations tested (Figure 2A and Appendix A). Interestingly, Fraction 3B lowered the A549 cell viability in a dose-dependent manner and did not exhibited cytotoxicity on human normal lung fibroblasts (WI38) (Figure 2B).

To establish the cell death signaling pathway induced by the active Fraction 3B, gene expression of both cell lines treated with 0.4 µg·mL^−1^ of Fraction 3B for 2 h was analyzed. The exposure time of 2 h was selected after having verified that cell death pathways were already expressed and activated after this time.

Control housekeeping genes for real-time qPCR were actin-beta (ACTB), beta-2-microglobulin (B2M), hypoxanthine phosphoribosyltransferase (HPRT1) and large ribosomal protein P0 (RPLP0). Figure 3 shows the relative expression ratios of the analyzed genes with respect to controls. Only expression values greater than a two-fold difference with respect to the controls were considered significant. The 2 h-treatment on A549 cells with Fraction 3B (Figure 3A) induced a strong up-regulation of the autophagy-related protein 12/gene (Atg12, 12.2-fold change) with a consequent increase in ATPase H+ Transporting V1 Subunit G2 (ATP6V1G2, 35.5-fold change) and BCL2/Adenovirus E1B 19kDa Interacting Protein 3 (BNIP3, 2.4-fold change). Two genes involved in mitophagy were significantly up-regulated: PTEN induced putative kinase 1 (PINK1, 3.6-fold change) and Parkin gene (11-fold change). Moreover, an up-regulation of the mitofusin-1/2 gene (MFN-1/2, 6.6-fold change) and the Voltage Dependent Anion Channel gene (VDAC, 5.9-fold change) was revealed. Activation of these genes caused a strong up-regulation of the Neighbor of BRCA1 gene 1 (NBR1, 12.6-fold change) and NIX gene also known as BCL2/Adenovirus E1B 19 kDa Interacting Protein 3-Like (5.9-fold change). The autophagy receptor Sequestosome 1 (SQSTMS1, 13.3-fold change) was strongly up-regulated indicating the degradation of intracellular vesicles.

Conversely, the same analysis carried out on the WI38 cells treated with Fraction 3B did not show any significant variation. Mitophagic genes in normal cells were not activated by the treatment, confirming that the active fraction did not induce cell death (Figure 3B).

Statistical analysis (ANOVA, *t*-student and Sidàk test; Figure 3C–E) performed on the gene expression data validated the significant differences of the biological activity of Fraction 3B on the A549 cells vs. WI38 cells.

## 3. Discussion

In this study, we investigated the cytotoxic activity of the dinoflagellate *Alexandrium minutum* specific against human lung adenocarcinoma cells A549. In particular, we use an innovative method of extraction by TRI Reagent^®^ that led to enrichment of cytotoxic macromolecules. Fractionation by hydrophobic column (Chromabond^®^ HR-X) followed by sequential steps of exclusion membranes increased the activity from IC_50_ = 1.3 µg·mL^−1^ in the extract to IC_50_ 0.4 µg·mL^−1^ in the fraction retained above 10 kDa (Fraction 3B). Chemical analysis of this material is consistent with the presence of one or more molecules with a molecular weight at least of 20 kDa and composed by almost 96% of carbohydrates and only 4% of protein. Differential gene expression analysis of A549 cells after treatment with the active Fraction 3B suggested an up-regulation of the genes involved in the mitochondrial autophagy (mitophagy) cell death pathway.

First, the gene ATG12 belonging to the autophagy-related gene family and involved in the first steps of formation and elongation of autophagosomes is activated. Then, ATP6V1G2, a downstream factor acting after ATG genes and responsible of the autophagosome degradation, is up-regulated [31]. Other genes, i.e., BNIP3 and NIX, are up-regulated by the active fraction treatment. These genes are probably involved in the connection between autophagy and cell death, even though the molecular mechanisms of BNIP3/NIX genes and connection with cell death are not well understood. Indeed, some studies report the activation of necrotic cell death by BNIP3/NIX genes, without the involvement of apaf-1, caspase 9 or 3 and cytochrome [32,33].

Confirmation of the mitophagic gene activation by Fraction 3B comes from up-regulation of Pink and Parkin genes. PINK1 is considered an upstream regulator of Parkin function, since the recruitment of Parkin to impaired mitochondria requires PINK1 expression and its kinase activity [34,35,36,37]. The change in membrane potential is the first signal for the mitophagic pathway through PINK1/parkin cascade [38,39]. PINK1, together with parkin protein, is responsible for the Mfn1/2 ubiquitination, playing a key role in the autophagic degradation of dysfunctional mitochondria (mitophagy) [40]. During the elimination of defective mitochondria, PINK1 mediates translocation of Parkin from the cytosol to mitochondria by an unknown mechanism. Sun et al. [41] reported that the three most abundant interacting proteins were the voltage-dependent anion channels (VDACs), pore-forming proteins in the outer mitochondrial membrane demonstrating their role as mitochondrial docking sites to recruit Parkin from the cytosol to defective mitochondria. Fraction 3B also up-regulated the gene encoding for VDAC in A549 cells. Moreover, the up-regulation of NBR1, a functional homolog of SQSTM1 (Sequestosome 1) with a role in PARK2-mediated mitophagy, indicates mitochondrial degradation [42].

These data predict that an extensive mitophagy may induce mitochondrial dysfunction and degradation (such as reduced mitochondrial membrane potential) with a consequential irreversible cell death [43]. Autophagy, as well as other peculiar death pathways, such as pyroptosis [44], are desirable targets for cancer therapy, since a new trend in anticancer research has arisen focusing on the discovery of new natural drugs that induce specific programmed cell death and ensure an optimal release of immunostimulatory signals [45], able to act as co-adjuvant stimuli for the reinforcement of the anti-cancer effect.

Until now, no natural products from marine microalgae are reported as inducers of mitophagic cell death in human tumor cells, although other studies show that compounds extracted from marine organisms are able to fight cancer cells, activating this specific mechanism. For instance, Kahalalide F (KF) (C75H124N14O16), a natural depsipeptides isolated from the Hawaiian herbivorous marine mollusk *Elysia rufescens*, activates mitochondrial autophagy in vitro in human prostate cancer cells [46].

This study is the first report that a water soluble high molecular weight compound isolated from the marine dinoflagellate, *Alexandrium minutum*, already known for toxins production, is able to up-regulate genes involved in a specific cell death in human lung adenocarcinoma cells through mitophagy activation, without affecting normal cell viability (human normal lung fibroblasts WI38). This mitophagic gene expression described in this study is likely attributed to a glycopeptide. The isolated fraction is much more active compared to the total extract, thus confirming the enrichment of the active fraction along with the chromatographic purification. Macromolecules based on polysaccharide and protein structure have already been described in other microalgae, including dinoflagellate [47]. These compounds have been associated to allelopathic activity by outcompeting other photoautotrophic species through growth inhibition. A large non-proteinaceous toxin with cytotoxic activity has been also reported from the dinoflagellate *Alexandrium tamarense* [48]. Although we have no direct proof that the cytotoxic molecule(s) of *A. minutum* belong to the same family, the indirect chemical evidence points out to this direction.

Further chemical studies are needed to better characterize the glycopeptide, a promising molecule with application for human health. Moreover, new chemical information about this molecule will help in elucidating the pathway for its synthesis, a crucial requirement for its biotechnological production. The latter might be pursued chemically or biologically, with for instance the improvement of its synthesis in *A. minutum* in modulating the growth-environmental conditions.

## 4. Materials and Methods

### 4.1. General Experimental Procedures

NMR spectra were recorded on a Bruker Avance DRX 600 equipped with a cryoprobe operating at 600 MHz for proton in D_2_O solution containing 1 mM sodium 3-trimethylsilyl [2,2,3,3-D_4_] propionate as a chemical shift reference for ^1^H spectra. Water was deionized and purified (Milli-Q, Millipore, Germany), whereas TRI Reagent^®^ and all solvents were purchased from Sigma (Aldrich, Milan, Italy). Chromabond^®^ HR-X resin was obtained from Macherey-Nagel GmbH (Düren, Germany). Ultrafiltration was carried out by Vivaspin 500 centrifugal concentrators (Sigma-Aldrich) with 3, 10 and 30 kDa molecular weight cut-off membrane of polyethersulfone (PES). Electrophoresis (SDS-PAGE) was performed by a mini gel apparatus BioRAD (Milan, Italy). To estimate the molecular mass, Precision Plus Protein™ Dual Color Standards (BioRad, Hercules, CA, USA) with 10 recombinant proteins of precise molecular weights (10–250 kD) were used.

### 4.2. Biological Material

*Alexandrium minutum*, isolated from the Gulf of Naples, was cultured in K medium [49], prepared with filtered (0.22 µm) natural sterile seawater, maintained at room temperature (22 ± 1 °C), under a 12:12 light/dark regime at 100 μmol m^−2^·s^−1^, in a 10 L sterilized polycarbonate carboy. The culture was harvested at the end of the stationary phase by centrifugation in a swing-out Allegra X12R (Beckman Coulter Inc., Palo Alto, CA, USA) at 2300× *g*, 4 °C, for 10 min. Cell growth was estimated by daily cells counts using a Bürker counting chamber (Merck, Leuven, Belgium) (depth 0.100 mm) under an inverted microscope. The harvested cell pellet (wet weight 5.7 g) was stored at −80 °C until analysis.

### 4.3. Extraction of A. minutum

According to manufacturer’s instructions, cells of *A. minutum* (5.7 g) were lysed in 45.6 mL TRI Reagent ^®^, which is a homogeneous water mixture of guanidine thiocyanate and phenol, for extraction of RNA, DNA and protein. The homogenate was centrifuged at 12,000× *g* for 10 min at 4 °C and the supernatant was added to 9.12 mL of CHCl_3_. The sample was vortexed and, after 15 min at room temperature, was centrifuged at 12,000× *g* for 15 min at 4 °C to separate the mixture into three phases including a red organic phase, an interphase and a colorless upper aqueous phase. The aqueous phase was transferred to a fresh tube and 31.9 mL EtOH was added to the sample. The suspension was centrifuged at 2000× *g* for 5 min at 4 °C and the supernatant was diluted with 114 mL acetone. After centrifugation at 12,000× *g* for 10 min at 4 °C, the pellet was suspended with MilliQ water and the resulting suspension was further centrifuged at 10,000× *g* for 10 min at 4 °C. The clear supernatant was recovered and lyophilized to give 350 mg of dry material.

### 4.4. Fractionation of the Hydrophilic Extract of A. minutum

The lyophilized extract of *A. minutum* was fractionated by HR-X ^®^ column according to a modified protocol of the method previously reported for the isolation of marine natural products [50]. Briefly, 3.8 g of Chromabond ^®^ HR-X resin was loaded into a glass column with methanol and equilibrated with MilliQ water. The dry extract of *A. minutum* (350 mg) was suspended in a minimal volume of MilliQ water and loaded onto the column. Fractionation was achieved by a four-step elution (see Appendix A) with 140 mL H_2_O (Fraction 1A, 340 mg), 110 mL ACN/H_2_O 7:3 (Fraction 1B, 5.3 mg), 85 mL acetonitrile (Fraction 1C, 0.8 mg), and 85 mL dichloromethane/MeOH 9:1 (Fraction 1D, 0.5 mg).

### 4.5. Ultrafiltration of the Active Fractions

Fraction 1B was dissolved in 2 mL of water and subjected to two sequential steps of ultracentrifugation on Vivaspin filters with cut-off of 3 and 10 kDa at 5000× *g* for 10 min at 24 °C. The fraction retained above 10 kDa was lyophilized to give 0.7 mg of the active Fraction 3B. Further ultrafiltration of this material with membrane of 30 kDa cut-off did not produce any useful separation of the activity.

### 4.6. Sizing of the Active Macromolecule by Diffusion NMR Analysis

The active Fraction 1B (5.3 mg) was dissolved in 700 µL of D_2_O with sodium 3-trimethylsilyl [2,2,3,3-D_4_] propionate for calibration. Diffusion edited ^1^H NMR spectra (1D-DOSY) were acquired with the pulse sequence using stimulated echo and 1 spoil gradient strength, 16 scans, big delta 0.1 s and little delta 2 ms and a gradient (gpz6) between 1% and 100%. The fitting of the diffusion dimension in the 2D-DOSY spectra was achieved using AU-program dosy to calculate gradient-diff ramp processing in Bruker TopSpin 3.5 [51].

### 4.7. Hydrolysis of the Active Fraction

The active Fraction 3B (0.3 mg) was dissolved in 200 µL of water and reacted with an equal volume of 2 M TFA at 100 °C for 4 h. The reaction mixture was dried under vacuum and the residue was dissolved in 1 mL of water. This solution was loaded on a Vivaspin membrane with 3 KDa molecular weight cut-off and centrifuged at 5000× *g* for 10 min at 24 °C. The filtrate was then analyzed by high-performance anion-exchange chromatography (HPAEC) using a Dionex LC30 (Dionex, Sunnyvale, CA, USA) for the analysis of sugars.

### 4.8. SDS-Polyacrylamide Gel Electrophoresis (SDS-PAGE)

Fractionation obtained by cut-off partitions, as described above, were analyzed by electrophoresis (SDS-PAGE) on 12% gels (1 mm) according to the method of Laemmli [52], loading 10 µg of each samples. After electrophoresis, the gels were detected by silver nitrate. Precision Plus Protein Dual Color Standards (BioRad) were used for molecular weight estimation.

### 4.9. Treatment of Human Cells

The human lung adenocarcinoma cell line A549 was purchased from the American Type Culture Collection (ATCC^®^ CCL185™) and grown in DMEM-F12 (Dulbecco’s modified Eagle’s medium) supplemented with 10% fetal bovine serum (FBS), 100 units mL^−1^ penicillin and 100 µg·mL^−1^ streptomycin in a 5% CO_2_ atmosphere at 37 °C. Human normal lung fibroblasts WI38 were grown in MEM (Eagle’s minimal essential medium) supplemented with 10% fetal bovine serum (FBS), 100 units mL^−1^ penicillin, 100 µg·mL^−1^ streptomycin, 2 mM of L-glutamine and non-essential amino acids (NEAA, 2 mM) in a 5% CO_2_ atmosphere at 37 °C. A549 and WI38 cells (2 × 10^3^ cells·well^−1^) were seeded in a 96-well plates and kept overnight for attachment. Chemical fractions were dissolved in dimethyl sulfoxide (DMSO) and used for the treatment of cells. The final concentration of DMSO used was 1% (*v*/*v*) for each treatment. Eighty percent confluent cells were treated in triplicate with fractions at 0.1, 1 and 10 μg mL^−1^ for 24 and 48 h in complete cell medium. Control cells were incubated with complete cell medium with 1% of DMSO.

### 4.10. Cell Viability

The antiproliferative effect of chemical fractions on cell viability was evaluated using the 3-(4,5-Dimethylthiazol-2-yl)-2,5-diphenyl tetrazolium bromide (MTT) assay (Applichem A2231) according to Gerlier et al. [53]. A549 and WI38 cells, after treatment with chemical fractions, were incubated with 10 µL (5 mg mL^−1^) of MTT for 3 h at 37 °C in a 5% CO_2_ atmosphere. Isopropanol (100 µL) was used to stop the incubation time and to solubilize purple crystals formed in each well by only viable cells. The absorbance was recorded on a microplate reader at a wavelength of 570 nm (Multiskan FC, THERMO SCIENTIFIC, Waltham, MA, USA). The effect of the fractions at different concentrations was reported as percent of cell viability calculated as the ratio between mean absorbance of each treatment and mean absorbance of control cells (A549 and WI38 cell lines treated with only 1% of DMSO).

### 4.11. RNA Extraction and Real-Time PCR

A549 (2 × 10^6^) and WI38 (2 × 10^6^) cells used for RNA extraction and analysis were seeded in Petri dishes (100 mm diameter) and kept overnight for attachment. After 2 h of treatment with 0.4 µg·mL^−1^ with Fraction 3B, cells were washed directly in the Petri dish by adding Phosphate-Buffered Saline (PBS) and rocking gently. Cells grown in complete medium without any treatment constituted experimental control. Cells were lysed in the Petri dish by adding 1 mL of Trisure Reagent (Bioline, Galgagnano, Lodi, Italy cat. BIO-38033). RNA was isolated according to the manufacturer’s protocol. RNA concentration and purity were assessed using the nanophotomer NanodroP (Euroclone, Pero, Milan, Italy). About 200 ng RNA was subjected to reverse transcription reaction using the RT^2^ first strand kit (Qiagen, Hilden, Germany, cat.330401) according to the manufacturer’s instructions. The qRT-PCR analysis was performed in triplicate using the RT^2^ Profiler PCR Array kit (Qiagen, cat.330231) to analyze the expression of genes involved in cell death signaling pathways. Plates were run on a ViiA7 (Applied Biosystems, Foster City, CA, USA, 384-well blocks) using a Standard Fast PCR Cycling protocol with 10 µl reaction volumes. Cycling conditions used were: 1 cycle initiation at 95 °C for 10 min followed by amplification for 40 cycles at 95 °C for 15 s and 60 °C for 1 min. Amplification data were collected via ViiA 7 RUO Software (Applied Biosystems). The cycle threshold (Ct)-values were analyzed with PCR array data analysis online software (GeneGlobe Data Analysis Center http://pcrdataanalysis.sabiosciences.com/pcr/arrayanalysis.php, Qiagen). Real time data were expressed as fold expression, describing the changes in gene expression between treated cells and untreated cells (control).

### 4.12. Statistical Analysis

One-way ANOVA was used for the assessment of variance within the control and treated groups and between these two experimental groups. Šidák method was applied to gene expression data to counteract the problem of multiple comparisons and to control the familywise error rate, pinpointing confidence intervals. *T*-test analysis determined statistical difference between means of the two experimental groups. Gene expression data were analyzed by PCR array data analysis online software (http://pcrdataanalysis.sabiosciences.com/pcr/arrayanalysis.php, Qiagen^®^). Only gene expression values greater than a 2.0-fold difference with respect to the controls were considered significant.

## Figures and Tables

**Figure 1 marinedrugs-16-00502-f001:**
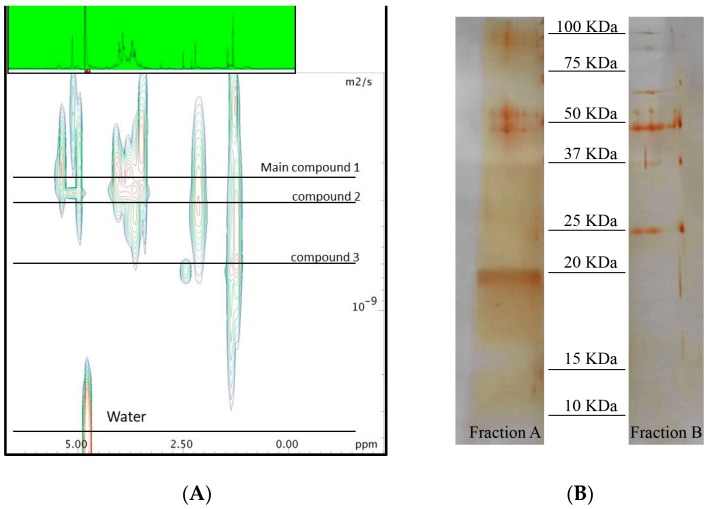
(**A**) 2D-Diffusion Ordered Spectroscopy (DOSY) spectra recorded in D_2_O at 600 MHz of Fraction 1B; and (**B**) Electrophoresis gel of Fractions 3B (active sample) and 4B (deglycosylated Fraction 3B sample).

**Figure 2 marinedrugs-16-00502-f002:**
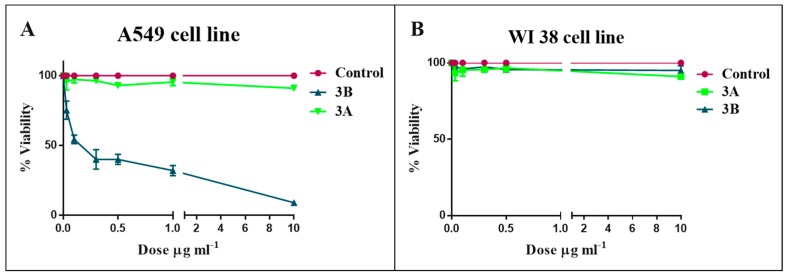
Effect of Fractions 3A (<10 KDa) and 3B (>10 KDa) on cell viability of human lung adenocarcinoma cells of (**A**) A549 and human normal lung fibroblasts and (**B**) WI38. Values are reported as mean ± S.D. compared to controls (100% viability) of three independent experiments. Concentrations tested were 0.1, 1 and 10 µg·mL^−1^ for 48 h.

**Figure 3 marinedrugs-16-00502-f003:**
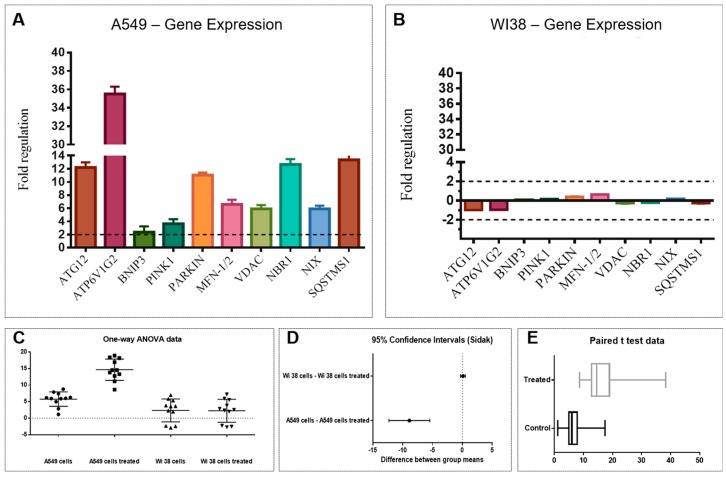
Effect of Fraction 3B on the expression levels of target genes in: human lung adenocarcinoma cells (A549) (**A**); and human normal lung fibroblasts (WI38) (**B**). All experiments were performed with RNA extracted from three different biological replicates and error bars represent ±S.D. Statistical analyses on the results obtained in (**A**,**B**); (**C**) One-way ANOVA); (**D**) Sidak; and (**E**) student’s t-test.

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
