# Peer review of "The Marine Dinoflagellate Alexandrium minutum Activates a Mitophagic Pathway in Human Lung Cancer Cells"

_marinedrugs, 2018, doi:10.3390/md16120502_

Reviewer 1 Report

The manuscript by Galasso et al. reports the role of toxic dinoflagellate Alexandrium minutum as source of water-soluble compounds with activity against human cancer cells. In particular the authors suggest that the mechanism of action of active fraction is death by mitophagy.

In my opinion the paper do not demostrate the conclusion. the mitophagy and cell death is not demonstrate. I invite you to deeper the subject matter and to resubmit the paper. 

Author Response

The manuscript by Galasso et al. reports the role of toxic dinoflagellate Alexandrium minutum as source of water-soluble compounds with activity against human cancer cells. In particular the authors suggest that the mechanism of action of active fraction is death by mitophagy.

In my opinion the paper do not demostrate the conclusion. the mitophagy and cell death is not demonstrate. I invite you to deeper the subject matter and to resubmit the paper.

We thank reviewer for his/her comment. We have now modified the discussion accordingly to the comment in order to better clarify our statement. As rightly commented by the reviewer, we did not univocally demonstrate the transduction of mitophagic pathway but we assessed the up-regulation of some genes that encode for factors involved in mitophagy on A549 adenocarcinoma cell line. We also demonstrated that the active component extracted from Alexandrium minutum did not affect autophagic pathway’ genes in normal cell line (WI38). These preliminary evidences can be the benchmark for further investigations about the biotechnological application of marine dinoflagellates for chemoprevention and/or cancer diseases treatments.

Reviewer 2 Report

In general the Manuscript entitle “The marine dinoflagellate Alexandrium minutum activates a mitophagic pathway in human cancer cells” is well designed and with novelty information regarding the effect of Alexandrium minutum in cancer cells.

Nevertheless, there are some issues that should be addressed:

·       In the title, the authors should specify the type of cancer (the effect of Alexandrium minutum could not be verified for all cancer cells). Therefore, correct for “…pathway in human lung cancer cells”.

·       In Results section, in 2.2 section, line 110, the authors mentioned the IC50=1.3 μl/ml of fraction 1B. Is this a result in “data not shown”? Please add the graph of the cytotoxicity effect of fraction 1B in A549 cells.

·       In Results section, in 2.2 section, line 112, the authors refer the IC50 of fraction 2B (IC50=0.8 μl/ml), however this result is not in Figure 2A, as point it out. Please add this data. In addition, the authors did not mention the fact that fraction 3A do not reach the IC50 value. Please add also this information in the text.

·       The Figures 2 shows the cell viability measured by MTT assay. The authors only used 3 concentrations to do the graph (the minimum required to do a dose response curve). Nevertheless, in order to have a reliable curve to determine the IC50 value, the authors should have used 5 different concentrations.

·       In Figure 3A, please change the axis Y for values of 2 in 2, in order to better visualize the 2 fold change, and add a dash (as in Figure 2B) in the value 2.

·       In Discussion section, line 157, the authors refer an IC50 of 1.4 μl/ml. Where it comes this value? Please correct/change this sentence.

·       In material and methods, section 4.9, line 283, please change the name of the cell line for “The human lung adenocarcinoma epithelial cell line A549...” in order to standardize the name of the cancer cell line along the manuscript.

·       In material and methods, section 4.9, line 292, the authors mentioned that they dissolved the fractions in DMSO at 1%. Please clarify if the control used in Figure 2 represents DMSO at 1%. The ideal concentration of DMSO in the cells is below 0.5% (normally range from 0.1% to 1%). Toxic effects on cells have been reported with 1% and higher percentage of DMSO. Please mention if the control used in all the experiments is DMSO at 1%, in order to exclude the toxic effect of the DMSO, and thus, being the cytotoxic effect observed in the cells only related to the fractions of Alexandrium minutum.

Author Response

Comments and Suggestions for Authors

In general the Manuscript entitle “The marine dinoflagellate Alexandrium minutum activates a mitophagic pathway in human cancer cells” is well designed and with novelty information regarding the effect of Alexandrium minutum in cancer cells.

Nevertheless, there are some issues that should be addressed:

In the title, the authors should specify the type of cancer (the effect of Alexandrium minutum could not be verified for all cancer cells). Therefore, correct for “…pathway in human lung cancer cells”.

We thank the reviewer for this comment. We modified the title of the manuscript as suggested.

In Results section, in 2.2 section, line 110, the authors mentioned the IC50=1.3 μl/ml of fraction 1B. Is this a result in “data not shown”? Please add the graph of the cytotoxicity effect of fraction 1B in A549 cells.

We kindly thank the reviewer for her/his attention in revising of our manuscript. As suggested, We added the graph (Figure S4) about the cytotoxicity of the fraction 1B in A549 cells in supplementary information.

In Results section, in 2.2 section, line 112, the authors refer the IC50 of fraction 2B (IC50=0.8 μg/ml), however this result is not in Figure 2A, as point it out. Please add this data. In addition, the authors did not mention the fact that fraction 3A do not reach the IC50 value. Please add also this information in the text.

We thank the reviewer for his/her comment. We added a figure in supplementary information (Figure S5) showing the results obtained with the fraction 2B for comparison with the fraction 3A (Figure 2A). As the reviewer suggested, we specified in the results section that fraction 3A did not affect cell viability.

The Figures 2 shows the cell viability measured by MTT assay. The authors only used 3 concentrations to do the graph (the minimum required to do a dose response curve). Nevertheless, in order to have a reliable curve to determine the IC50 value, the authors should have used 5 different concentrations.

We thank the reviewer for this comment; we added the data regarding the other concentrations tested on A549 and Wi38 cell lines. The estimation of IC50 was made considering the 5 concentrations.

In Figure 3A, please change the axis Y for values of 2 in 2, in order to better visualize the 2 fold change, and add a dash (as in Figure 2B) in the value 2.

We agree with the reviewer and we modified the figure by adding a dash to highlight the range between -2 and 2 in fold change regulation.

In Discussion section, line 157, the authors refer an IC50 of 1.4 μl/ml. Where it comes this value? Please correct/change this sentence.

We thank the reviewer for this comment allowing us to correct this value reported wrongly since it is referred to IC50 of fraction 1B (1.3 µg ml-1).

In material and methods, section 4.9, line 283, please change the name of the cell line for “The human lung adenocarcinoma epithelial cell line A549...” in order to standardize the name of the cancer cell line along the manuscript.

We thank the reviewer for this comment; we standardized the name of both cell lines. In the revised version: A549 is “human lung adenocarcinoma cells” and to WI38 is “human normal lung fibroblasts”.

In material and methods, section 4.9, line 292, the authors mentioned that they dissolved the fractions in DMSO at 1%. Please clarify if the control used in Figure 2 represents DMSO at 1%. The ideal concentration of DMSO in the cells is below 0.5% (normally range from 0.1% to 1%). Toxic effects on cells have been reported with 1% and higher percentage of DMSO. Please mention if the control used in all the experiments is DMSO at 1%, in order to exclude the toxic effect of the DMSO, and thus, being the cytotoxic effect observed in the cells only related to the fractions of Alexandrium minutum.

We agree with the reviewer. The control group correspond to cells grow in the multi-well plate used for the treatments and incubated with complete cell medium with the same percentage of DMSO used to dissolve samples (1%). Thus, data of percentage of cell viability are compared to control (which represents the 100% of viability). We clarified this aspect in the revised version of the manuscript, pag.8:

section 4.9 “Control cells were incubated with complete cell medium with 1% of DMSO” and

section 4.10: “The effect of the fractions at different concentrations was reported as percent of cell viability calculated as the ratio between mean absorbance of each treatment and mean absorbance of control cells (A549 and WI38 treated with only 1% of DMSO)”

Reviewer 3 Report

The article entitled "The marine dinoflagellate Alexandrium minutum  activates a mitophagic pathway in human cancer cells " presents the cytotoxic potential of extract from marine dinoflagellate Alexandrium minutum. But the manuscript has insufficient research contents and findings so as to maintain the merit to be considered for publication in "Marine drug".

The active component in the bioactive fraction is not properly identified, hence I would recommend for proper separation, purification and structural elucidation, to estimate the exact chemical constituent.

The author only checked activity against A549 cells. I would recommend for checking activity against different cell lines so that they can assess the comparative effect on different cell lines as lung, gastric, brian, etc. The cell line with greatest cytotoxity can be taken for further studies.

The qRT PCR is not sufficient for assessing the differential expressions profiling by the effect of the compound. Hence, I would recommend to support the findings by Western blot analysis of appropriate cancer marker proteins.  

The result can be further supported by the experiments in the appropriate mouse models.

Author Response

The article entitled "The marine dinoflagellate Alexandrium minutum  activates a mitophagic pathway in human cancer cells " presents the cytotoxic potential of extract from marine dinoflagellate Alexandrium minutum. But the manuscript has insufficient research contents and findings so as to maintain the merit to be considered for publication in "Marine drug".

The active component in the bioactive fraction is not properly identified, hence I would recommend for proper separation, purification and structural elucidation, to estimate the exact chemical constituent.

We agree with this reviewer but we respectfully notice that we never claimed the structure characterization of the bioactive component in the manuscript. In fact, as the reviewer writes, the identification of the molecule is far from being complete. Nevertheless, the results reported in the manuscript are rather clear about to define a water-soluble biopolymer as specific trigger of mitophagy in human lung adenocarcinoma cells. As suggested, we are attempting identification of this molecule but this is out of the scope of the present manuscript. We hope that this reviewer agrees with us on the fact that, albeit important, the elucidation of the structure of the active molecule does not change the intrinsic value of the current manuscript.

The author only checked activity against A549 cells. I would recommend for checking activity against different cell lines so that they can assess the comparative effect on different cell lines as lung, gastric, brain, etc. The cell line with greatest cytotoxity can be taken for further studies.

We thank the reviewer for this comment. In order to choose the active fraction and the best cell line to treat for further chemical analysis, we compared 3 different cell lines using a large range of concentrations.

We added these results, which were lacking in the previous version, in the supplementary information (Figure S4) of this revised version of the ms.

The qRT PCR is not sufficient for assessing the differential expressions profiling by the effect of the compound. Hence, I would recommend to support the findings by Western blot analysis of appropriate cancer marker proteins. 

We thank the reviewer for his/her comment. The aim of the study was a first attempt to depict the possible cell signalling pathway triggered by the bioactive component at molecular level measuring gene expression variations both on cancer and normal cell lines of the same tissue origin (lung).

We modified the discussion in order to clarify this aspect, i.e. providing information about the most probable pathway activated in order to address future investigations on the pure molecule/s responsible for the observed bioactivity and its pharmakinetics mechanisms, for instance using in vivo models.

The result can be further supported by the experiments in the appropriate mouse models.

We thank the reviewer for his/her suggestion. As anticipated in the previous answer, further investigations will address this aspect; we wish to deeply continue the investigation about Alexandrium minutum as potential source of anticancer compounds also using mouse models for the assessing of bioactivity observed in vitro.

Reviewer 4 Report

The manuscript by Galasso et al. describes the isolation and initial characterization of anticancer compounds from the marine dinoflagellate Alexandrium minutum. Specifically, a bioactive fraction containing 96% carbohydrates and 8% protein was isolated from A. minutum after iterative rounds of purification. The active compound is thought to be a glycoprotein, and proved to be active against human lung cancer cells (IC50= 0.4 mg/mL). Differential gene expression suggested that the glycoprotein activates mitophagy. Importantly, the glycoprotein was not active against, and mitophagy was not observed in, normal lung cells. The authors believe that this is the first report of mitophagy activated by a marine dinoflagellate.

This work is suitable for publication in Marine Drugs after consideration of the following minor points:

1. There are minor typos and grammatical errors throughout the manuscript.

2. Figure 1B should be clarified. The ladder overwhelms the bands of interest in the isolated fractions.

3. This reviewer was not provided the Supporting Information.

Overall, this reviewer feels that this manuscript is very interesting and well written. While further characterization of the glycoprotein, including sequencing, BLAST comparison with other dinoflagellates/marine organisms, structural information, etc. would be of great interest, this reviewer recognizes that that is beyond the scope of this initial paper.

Author Response

The manuscript by Galasso et al. describes the isolation and initial characterization of anticancer compounds from the marine dinoflagellate Alexandrium minutum. Specifically, a bioactive fraction containing 96% carbohydrates and 8% protein was isolated from A. minutum after iterative rounds of purification. The active compound is thought to be a glycoprotein, and proved to be active against human lung cancer cells (IC50= 0.4 mg/mL). Differential gene expression suggested that the glycoprotein activates mitophagy. Importantly, the glycoprotein was not active against, and mitophagy was not observed in, normal lung cells. The authors believe that this is the first report of mitophagy activated by a marine dinoflagellate

This work is suitable for publication in Marine Drugs after consideration of the following minor points:

There are minor typos and grammatical errors throughout the manuscript.

We thank the reviewer for this comment. All authors have read and corrected grammatical errors improving English editing.

Figure 1B should be clarified. The ladder overwhelms the bands of interest in the isolated fractions.

We have resampled the imagine and improved the definition but we are afraid that the quality cannot be better than we showed in the revised file here attached. In fact, the protein content in the two fractions is very low thus it is not possible to increase the band intensity. However, we have changed the figure by reporting only the ladders related to the two fractions. An improved copy of original gel is reported in the Supporting Material as supplementary figure.

This reviewer was not provided the Supporting Information.

We are sorry for the lack of supporting information, and we did not understand why. We hope that he/she will get them in the new round of review.

Overall, this reviewer feels that this manuscript is very interesting and well written. While further characterization of the glycoprotein, including sequencing, BLAST comparison with other dinoflagellates/marine organisms, structural information, etc. would be of great interest, this reviewer recognizes that that is beyond the scope of this initial paper.

We thank the reviewer for this comment. Since, as already stated in answers to previous queries addressed by the reviewer 2, our goal is to pursue the investigations on this novel finding, we welcome any suggestions and collaborations in this way.

Round  2

Reviewer 1 Report

I accept their modifications.

Author Response

We thank the reviewer for his/her positive comment and for the time he/she dedicated to comment our ms, improving its quality.

Reviewer 3 Report

Dear authors,

1. The present form of article looks better but still I would like to recommend authors to depict probable candidates for activity. Is it possible to perform GC-MS and check against mass library? There are some useful platforms as https://gnps.ucsd.edu/ProteoSAFe/static/gnps-splash.jsp to predict some compounds by pattern of MS/MS fragmentation.

2. How do perform RT-PCR analysis? What were the controls? Did you use 2^-(del del ct)? Please explain the procedure in details or else provide appropriate references.

3. I would like to still recommend for confirmation of pathway by western blot. mRNA have very short life and there are many regulating RNAs, so protein level assessment is compulsory to depict the exact effect. 

Author Response

1. The present form of article looks better but still I would like to recommend authors to depict probable candidates for activity. Is it possible to perform GC-MS and check against mass library? There are some useful platforms as https://gnps.ucsd.edu/ProteoSAFe/static/gnps-splash.jsp to predict some compounds by pattern of MS/MS fragmentation.

We acknowledge the reviewer for his/her suggestion. Unfortunately, the polarity and high molecular weight of the molecule do not allow its separation and description using GC-MS. Our idea is to analyse it with liquid chromatography coupled with tandem mass spectrometry (LC-MS/MS) using a shotgun approach. However, to evaluate and suggest a chemical structure for this molecule, we need genomic information. This is a reason why we cannot perform this analysis but we are planning using these approaches when we will get a pure or semi-pure form of the molecule.

2. How do perform RT-PCR analysis? What were the controls? Did you use 2^-(del del ct)? Please explain the procedure in details or else provide appropriate references.

We thank the reviewer for this comment, highlighting that relevant information on real time procedure are lacking in the manuscript.

RT2 Profiler PCR Arrays are reliable tools for analysing the expression of a focused panel of genes. Specifically, we used the RT² Profiler™ PCR Array Human Cell Death Pathway Finder (Product n° 330231; Cat. n° PAHS-212Z).

The Human Cell Death Pathway Finder RT² Profiler PCR Array profiles the expression of 84 key genes with relevant role in the central mechanisms of cell death: apoptosis, autophagy, and necrosis. The output of this array gives insights into the central cell death mechanism(s).

All threshold cycle (CT) values obtained were analysed using ViiA™ 7 Software (Applied Biosystems) and the GeneGlobe Data Analysis Center. These tools are used for transforming real-time PCR CT values in “fold expression”, through Delta-Delta CT method (Fold expression = 2-∆∆Ct) to facilitate result interpretation. This approach needs housekeeping genes, which were the following: actin-beta (ACTB), beta-2-microglobulin (B2M), hypoxanthine phosphoribosyltransferase (HPRT1) and large ribosomal protein P0 (RPLP0).

The control of this analysis was the cells (A549 and WI38) grown in complete medium without any treatment.

The text of the ms was changed as follow:

-lines 326-327: “Cells grown in complete medium without any treatment constituted experimental control.”

-lines 336-338: “( GeneGlobe Data Analysis Center http://pcrdataanalysis.sabiosciences.com/pcr/arrayanalysis.php, Qiagen)”

-lines 338-340: “Real Time data were expressed as fold expression, describing the changes in gene expression between treated cells and untreated cells (control)”

3. I would like to still recommend for confirmation of pathway by western blot. mRNA have very short life and there are many regulating RNAs, so protein level assessment is compulsory to depict the exact effect.

We agree with the reviewer’s comment, modifying a little the discussion section. In this manuscript, our goal was to describe and share the results on a potential effect, mitophagy, on lung cancer cells exerted by a polar fraction extracted from Alexandrium minutum describing variations in term of gene expression after treatment. At this step, this important result recalls other investigations once we will be able to purify the compound, characterizing the biological activity at protein level by western blot and ELISA analysis. This will allow to monitor the mitophagic pathway, genetically observed/hypothesized.